# The relationship between social support, stressful events, and menopause symptoms

**Megan Arnot** *, **Emily H. Emmott, Ruth Mace**

Department of Anthropology, University College London, London, United Kingdom

* megan.arnot.13@ucl.ac.uk

## Abstract

Many women going through the menopausal transition experience vasomotor symptoms (VMS), and research has shown that there is a large amount of variation in their frequency and severity. Many lifestyle factors have been found to co-vary with VMS, including the level of social support received by the woman, and how stressed she is. Stress is well documented to worsen menopause symptoms, and there is some evidence that support eases them; however, there is little research into whether support is an effective buffer against the negative effects of stress on VMS. Using nine years of data from the Study of Women's Health Across the Nation (n = 2718), we use multilevel Poisson regression with random effects to test: 1) if more social support is associated with decreased VMS frequency, 2) if increased life stress worsens VMS, and 3) if support acts as a buffer against stress. After adjusting for age, marital status, smoking, self-perceived overall health, ethnicity, and menopausal status, we find that stress increases the frequency of VMS. Contrary to our hypothesis, we did not find strong evidence that emotional support led to lower VMS frequency, or that support buffers against the effects of stress. Experience of a stressful event, but not amount of social support, was included in the best fitting model; with the degree to which the woman was upset by the life stressor having the largest effect on menopause symptoms. Here, women who said they were currently upset by a stressful event experienced 21% more VMS than women who had experienced no life stressor. This research highlights that social factors may impact the menopausal transition.

## Introduction

Menopause is often a significant event for women; and, for many, the process is not smooth. Rather, it is accompanied by a multitude of physical symptoms, with vasomotor symptoms (VMS) being the most frequently reported, which include hot flashes, night sweats, and cold sweats [1]. In addition to VMS being generally bothersome, they have also been observed to have a negative impact on women's mental health and wellbeing [2]. Typically, menopause symptoms are confined to the perimenopausal period; however, they are known to occasionally persist beyond final menstruation [3]. They are thought to be primarily the result of hormonal fluctuations that occur during the menopausal transition, with higher levels of follicular stimulating hormone and lower oestradiol being associated with a greater likelihood of reporting VMS [4].

**Data Availability Statement:** All data are publicly available at https://www.icpsr.umich.edu/web/ICPSR/series/00253 with no restrictions. Data were cleaned for use within this study.

**Funding:** MA is funded by the ESRC-BBSRC Soc-B Centre for Doctoral Training (grant no. ES/P000347/1). The funders had no role in study design, data collection and analysis, decision to public, or preparation of the manuscript. https://www.ucl.ac.uk/soc-b-biosocial-doctoral-training/.

**Competing interests:** The authors have declared that no competing interests exist.

However, while all women experience these hormonal changes during the menopausal transition, not all report VMS [5]: while population estimates vary [6], it is thought that over 70% of all women experience VMS at some point during the menopausal transition [3,7], meaning the minority of women do not experience VMS. Further, even amongst the women who do report VMS, the symptomatology is not uniform within and between populations, with women reporting them to varying degrees of severity and frequency [3,8–13]. Given the negative impact of VMS reported by women [14], it is important to understand what co-varies with VMS prevalence and severity, as it may better equip women to deal with the menopausal transition.

In recent years, different social and ecological factors have been found to associate with VMS, including ethnicity, living arrangements, and socioeconomic position (SEP) [3,12,15]. Importantly, it has been argued that stress may mediate the relationship between such factors and the women's reported menopause symptoms [12,16]. Experience of stressors and increased perceived stress have been linked to higher levels of cortisol and fibrinogen, with the former being associated with the suppression of the immune, reproductive, and digestive systems, and the latter with increased inflammation [17]. As such, stress has causally been associated with a greater risk of cardiovascular disease [18], faster progression of HIV [19], delayed wound healing [20,21], decreased receptivity to the influenza vaccine [22], and a greater risk of upper respiratory tract infection [23,24]. Moreover, a number of studies have found that increased stress (both perceived stress and stressful experiences) can worsen the frequency and severity of menopause symptoms [3,25–28]; however, the causal pathway underlying this particular association is currently unclear [16].

The potential effect of stress on menopause symptoms is of importance to understand: midlife is often a highly stressful time for women, with many major life events coinciding with one another, such as parental death, children leaving home, and divorce [29–31]. Further, going through menopause can itself be a stressful event, as it marks a transition to a new phase of life, with many women reporting a sense of loss accompanying the end of their fertility [32]. Previous research suggests that a strong social network and higher levels of social support throughout the menopausal transition may reduce menopause symptom frequency and severity [3,33–35]. Similar associations have also been observed between social relationships and other health outcomes [36,37], with strong social ties being related to an increased likelihood of cancer recovery [38,39], a reduced risk of Alzheimer's and dementia [40,41], and also protect against cardiovascular disease [42,43]. Exactly how social ties work to sustain and/or improve health and well-being is not completely clear [37]. However, possible pathways include lower social support manifesting itself in more negative behavioural (i.e. poorer health behaviours) and psychological (i.e. depression) processes [36].

In addition to the noted direct benefits of social support, it is also thought that support can act as a buffer against the negative effects of stress on health [44]. Here, a stronger support system is thought to lessen the impact of stress, decrease its intensity, and also help to correct any maladaptive behaviours forming as a result of stress (e.g. alcoholism) [44]. Hence, social support may weaken the association between stress and negative health outcomes by moderating the actor's reaction or perception of the stress they are experiencing. Despite the fact that stress has been widely associated with worse menopause symptoms, to our knowledge, the stress-buffering hypothesis for social support has not been explicitly tested in relation to VMS.

In this study, we build on previous research into stress, support and menopause symptoms by testing whether social support acts as an effective buffer against the negative influence of stress on VMS. First, we test whether higher levels of social support and stress associate with less frequent VMS and more frequent VMS, respectively. Further, we test whether social support buffers against the proposed negative impact of stress on VMS. This research aims to clarify how support and stress are associated with one another in regards to VMS.

## Material and methods

### Study sample

Data from the Study of Women's Health Across the Nation (SWAN) were used in this study. SWAN is an ongoing, multi-site, longitudinal cohort study currently being carried out in the US. Data collection began in 1996/97, with women aged between 42 and 52 being recruited initially. Criteria to be part of the baseline cohort including having an intact uterus, at least one ovary, experience of a menstrual period in the past three months, and not currently being pregnant. 11 years' worth of data is currently publicly available for analysis [45–53], and this present study utilises data from nine years of interviews based on data available at the time of analysis (visits 1, 2, 3, 4, 5, 6, 8, 9 and 10). Baseline data (visit 0) was omitted as the stress measure was not comparable to the question patterning in later years, and data from visit 7 was also not included as participants were not asked any questions on social support in that year of interviews.

### Variables

**Vasomotor symptoms.** VMS served as the outcome variable in all analyses, and was measured through combining how often women experienced hot flashes, cold sweats and night sweats in the two weeks approaching the interview. These were measured individually on a Likert scale (not at all; 1–5 days; 6–8 days; 9–13 days; every day), and we then assigned each of these responses a value ranging from 0 to 5 (i.e. 0 = not at all; 1 = 1–5 days; and so on) and summed the woman's experience of the three symptoms. This left each respondent with a score between 0 and 15, in which a higher score indicated more frequent VMS.

**Social support.** SWAN provides four measures of social support which include how often the respondent feels she has someone to confide in, someone to listen to her, take her to the doctors, and help out when sick (see S1 Text Figure A). These were measured on a 5-point Likert scale (none of the time; a little of the time; some of the time; most of the time; all of the time), and taken from the Medical Outcome Study: Social Support Survey which is one of the most widely used instruments for measuring social support [54], and shows high levels of validity and reliability [55–58]. An average measure of support was created, in which the Likert scales for each measure of support were converted to numeric (i.e. none of the time = 1; a little of the time = 2, and so on), then summed and averaged to create a value ranging from 1–5, where a higher value indicated greater social support [54].

**Stress.** For this study, experience of stressful events was used as a metric of how stressed the participant was. SWAN asks participants about the occurrence of, and perceived stressfulness of, 18 life stresses that have occurred over the past year. These include the death of close kin, employment changes, and changes in household structure (see Fig 1 for the specific stressors and individual relationships). This is a modified version of the Psychiatric Epidemiology Research Interview life events scale [59], which has been validated across multiple racial and ethnic groups and shows high internal consistency [60]. Each stressful event was measured on a 5-point Likert scale which indicated whether she had experienced the event or not and how much it affected her (no–did not experience; yes–not upsetting; yes–somewhat upsetting; yes–very upsetting; yes–very upsetting and still upsetting). From these 18 questions, we derived two variables:

1. A 'stress index' which is the maximum amount of stress the participant experienced across all of the 18 stressful events. It can be seen in Fig 1 that all the events individually present a similar relationship with VMS, with increased upset being associated with more VMS, and therefore we created a composite measure from these which indicates the maximum amount of stress experienced by the respondent. For example, if a woman answered "No"

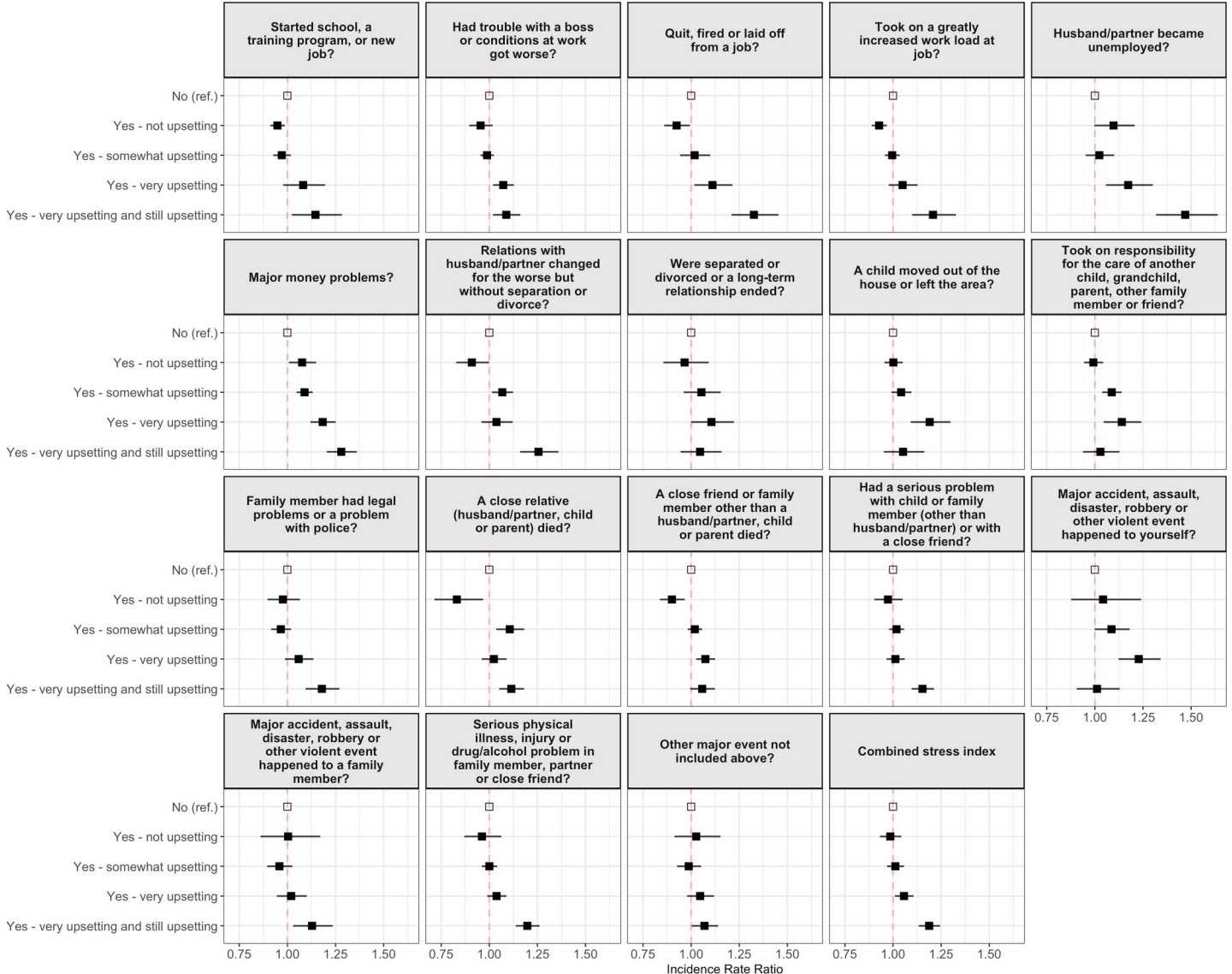

**Fig 1. Relationship between the 18 individual measures of stress and vasomotor symptoms.** Models are unadjusted, with a higher Incidence Rate Ratio indicating more frequent vasomotor symptoms.

to 17 of the events, and stated that one event had "Somewhat upset her", her stress index would be "Yes–somewhat upsetting". An outline of the derivation of this variable can be found in the S1 Text Table A. In summary, a woman's stress index can be seen as:

*Did the participant experience a stressful event over the past year, and was she upset by it?*

With the possible response being either: No; Yes, but she did not find it upsetting; Yes, and she found it somewhat upsetting; Yes, and she found it very upsetting, or; Yes, and she found it very upsetting at the time and is still upset by it.

2. A 'stress dose variable' was also created to measure the woman's 'stress dose', or how many stressors she has experienced. This was therefore a count variable which ranged

from 0 to 18, with a higher number indicating that she has experienced more stressful events. An explanation of how this variable was derived is presented in S1 Text Table B.

Both the stress index and stress dose variables were included as time varying. A detailed description of the creation of the variables and variable selection can be found in S1 Text (Tables A and B, and Figure C).

**Covariates.** Covariates were selected based on existing research into VMS and included as time varying. These included age, marital status (divorced/separated, married/in a relationship, widowed) [61], smoking habits (never smoked, ever smoked) [62], maximum level of education (less than high school, high school, some college/technical college, college degree, post-graduate degree) [63], ethnicity (self-identified as Black or African American, Chinese American, Japanese American, non-Hispanic Caucasian, Hispanic) [3], self-perceived overall health (poor, fair, good, very good, excellent) [64], and menopausal status (pre-menopausal, early perimenopause, late perimenopause, post-menopausal, and not menstruating for another reason). In all analyses, age was centred on the mean, with cubic and quadratic terms also included.

## Analyses

To test whether support and stress influence vasomotor symptom experience throughout the menopausal transition, multivariable, multilevel Poisson regression with mixed effects was used. This method was deemed as appropriate as the VMS in our data follows a Poisson distribution, with most women reporting infrequent symptoms. In total, 8 models were made, which are shown in Table 1. An initial model that comprised solely of the covariates (the *Base model*) was first created as a unit of comparison. Social support, the stress index, and stress dose variables were then subsequently added to the base model (the *Support model*, *Stress model 1* and *Stress model 2*, respectively) individually, and then together (*Full model 1* and *2*; stress measures were always included in separate models due to collinearity). Finally, to measure whether support buffers against stress, two models were made with an interaction effect between support and the two measures of stress (*Interaction model 1* and *2*).

Model fitting was then carried out using the Akaike Information Criterion (AIC), which has been shown to be appropriate for Poisson regression with random effects [65]. The AIC

**Table 1. Candidate models.**

| Model name | Covariates |
|---|---|
| Base model | Age + Age$^2$ + Age$^3$ + Marital status + Smoking habits + Education + Ethnicity + Overall health + Menopause status |
| Support model | Social support + Age + Age$^2$ + Age$^3$ + Marital status + Smoking habits + Education + Ethnicity + Overall health + Menopause status |
| Stress model 1 | Stress index + Age + Age$^2$ + Age$^3$ + Marital status + Smoking habits + Education + Ethnicity + Overall health + Menopause status |
| Stress model 2 | Stress dose + Age + Age$^2$ + Age$^3$ + Marital status + Smoking habits + Education + Ethnicity + Overall health + Menopause status |
| Full model 1 | Social support + Stress index + Age + Age$^2$ + Age$^3$ + Marital status + Smoking habits + Education + Ethnicity + Overall health + Menopause status |
| Full model 2 | Social support + Stress dose + Age + Age$^2$ + Age$^3$ + Marital status + Smoking habits + Education + Ethnicity + Overall health + Menopause status |
| Interaction model 1 | Social support*Stress index + Age + Age$^2$ + Age$^3$ + Marital status + Smoking habits + Education + Ethnicity + Overall health + Menopause status |
| Interaction model 2 | Social support*Stress dose + Age + Age$^2$ + Age$^3$ + Marital status + Smoking habits + Education + Ethnicity + Overall health + Menopause status |

shows which model is a relatively better fit to the data, and also penalises models for complexity. The model with the lowest AIC value is deemed to best fit the data, and a ΔAIC increase of more than 3 signifies weaker support for the model [66]. Models were then weighted based on how much their AIC value changed relative to the best fitting model [66]. All analyses were performed in *R* version 4.0.2 [67] using the packages *lme4* [68] and *ggeffects* [69]. Model fitting was carried out using *AICcmodavg* [70], and data visualisations created using *ggplot2* [71].

## Results

### Descriptive statistics

In the first round of interviews used in this analysis (n = 2718) women were aged between 42 and 54 (mean [M]: 46.91; standard deviation [SD]: 2.69), with the majority of participants either being married or in a relationship (76.09%). A greater proportion of women had never smoked (58.06%), and very few women were not educated to a high school standard, with most women having attended some kind of college or technical school (32.38%). Half of the women in the sample were white (49.71%), and at this point in the study very few women were post-menopausal (1.51%), with most of them being in early perimenopause (60.60%). Women were receiving high levels of social support on average (M: 4.18; SD: 0.78, see S1 Text Figure B), most had experienced a stressful event that upset them to some degree (76.49%), and the stress dose variable was positively skewed (S1 Text Figure D), with women experiencing on average 3 (SD: 2.45) stressful events in the past year. At this stage, the majority of women were not reporting frequent menopause symptoms (M: 1.21; SD: 2.05). Full descriptive statistics are presented in Table 2.

### Base model

The *Base model*, which comprised of all the covariates, was created to serve as a unit of comparison (see S1 Text Table C for full results). Within this model, women who were married experienced more frequent VMS compared to divorced or separated women (IRR: 1.10; 95% CI: 1.05–1.16), as did women who had ever smoked (IRR: 1.22; 95% CI: 1.12–1.32). There was a near linear relationship between level of education and reporting of menopause symptoms, with a longer time spent in education being associated with decreased VMS frequency (IRR: 0.72; 95% CI: 0.59–0.88). As in previous studies [3], African American reported VMS most often, with Japanese American women experiencing them least (IRR: 0.41; 95% CI: 0.35–0.48); and women who perceived themselves to be healthier also reported less frequent VMS (IRR: 0.65; 95% CI: 0.58–0.72). Women who were in late perimenopause (IRR: 1.87; 95% CI: 1.79–1.96) or had experienced menopause (IRR: 1.44; 95% CI: 1.37–1.51) reported symptoms more often.

### Support and stress

Our study is testing the prediction that social support and stress influence symptom severity, and that support would act as a buffer against the effects of stress. Should this be the case, we would expect models with these variables included to better fit the data, in addition to an interaction effect. Results from model fitting are presented in Table 3, where it can be seen that the best fitting model is *Stress model 1* (ΔAIC = 0.00), which included the stress index and covariates. The inclusion of social support (*Full model 1* ΔAIC = +0.66) and an interaction term (*Interaction model 1* ΔAIC = +16.13) worsened the model fit. The model that included just the covariates worst fit the data (*Base model* ΔAIC = +91.52).

**Table 2. Characteristics of the participants in visit 1.**

| Variable | n (%) | Mean (S.D) |
|---|---|---|
| Vasomotor symptoms | | 1.21 (2.05) |
| Social support | | 4.18 (0.78) |
| Stress index (experienced a stressful life event in past year) | | |
| No | 388 (14.3) | |
| Yes—not upsetting | 251 (9.2) | |
| Yes—somewhat upsetting | 766 (28.2) | |
| Yes—very upsetting | 642 (23.6) | |
| Yes—very upsetting and still upsetting | 671 (24.7) | |
| Stress dose | | 3.07 (2.45) |
| Age | | 46.91 (2.69) |
| Marital status | | |
| Divorced/Separated/Single | 601 (22.1) | |
| Married/In a relationship | 2068 (76.1) | |
| Widowed | 49 (1.8) | |
| Smoking habits | | |
| Ever smoked | 1140 (41.9) | |
| Never smoked | 1578 (58.1) | |
| Education | | |
| Less than high school | 169 (6.2) | |
| High school | 443 (16.3) | |
| Some college/technical school | 880 (32.4) | |
| College degree | 569 (20.9) | |
| Post-graduate education | 657 (24.2) | |
| Ethnicity | | |
| African American | 687 (25.3) | |
| Chinese American | 220 (8.1) | |
| Japanese American | 262 (9.6) | |
| White | 1351 (49.7) | |
| Hispanic | 198 (7.3) | |
| Self-perceived overall health | | |
| Poor | 50 (1.8) | |
| Fair | 294 (10.8) | |
| Good | 820 (30.2) | |
| Very good | 1073 (39.5) | |
| Excellent | 481 (17.7) | |
| Menopausal status | | |
| Pre-menopausal | 676 (24.9) | |
| Early peri | 1647 (60.6) | |
| Late peri | 112 (4.1) | |
| Menopaused | 41 (1.5) | |
| Other | 242 (8.9) | |

In the best fitting model (*Stress model 1*) it can be seen that the degree to which the woman was affected by the stressor was most critical. Fig 2 displays the predicted VMS at any given age based on experience of a stressor. Here, being still upset by a stressful event increased VMS frequency by 21% (95% CI: 1.15–1.26), and women who found the stressful event very upsetting at the time (but were no longer upset by it) experienced a 7% increase in VMS frequency

**Table 3. Results from model fitting based on Akaike Information Criterion (AIC) value.**

| Model | K | AIC | ΔAIC | $w_i$ |
|---|---|---|---|---|
| Stress model 1 | 28 | 64167.97 | 0.00 | 0.58 |
| Full model 1 | 29 | 64168.64 | 0.66 | 0.42 |
| Interaction model 1 | 29 | 64184.11 | 16.13 | 0.00 |
| Stress model 2 | 25 | 64187.53 | 19.55 | 0.00 |
| Full model 2 | 26 | 64187.96 | 19.99 | 0.00 |
| Interaction model 2 | 25 | 64204.12 | 36.14 | 0.00 |
| Support model | 25 | 64258.49 | 90.52 | 0.00 |
| Base model | 24 | 64259.50 | 91.52 | 0.00 |

A lower AIC value indicates better fit for the model, and a ΔAIC increase of more than 3 signifying weaker support for the model. $w_i$ shows the Akaike weights that indicate the level of support for the model, or the 'model probabilities'.

(IRR: 1.03–1.13). However, those who were not upset by the stressful event (IRR: 1.01; 95% CI: 0.95–1.06) or only found it somewhat upsetting (IRR: 1.03; 95% CI: 0.98–1.07) did not experience a statistically significant increase in VMS at the time of reporting (see Table 4).

As our current understanding of the physiological pathways around stress and VMS leads us to predict that current stress is an important determinant of VMS, we then also conducted an additional *post hoc* analysis with a time-lagged stress measure to test whether current stress is a better predictor of VMS than historical stress. Results (see S1 Text Table C) show that experience of stressful life events from a year prior to VMS reporting have a weaker relationship with VMS frequency than currently reported stressful events. Here, a very upsetting event

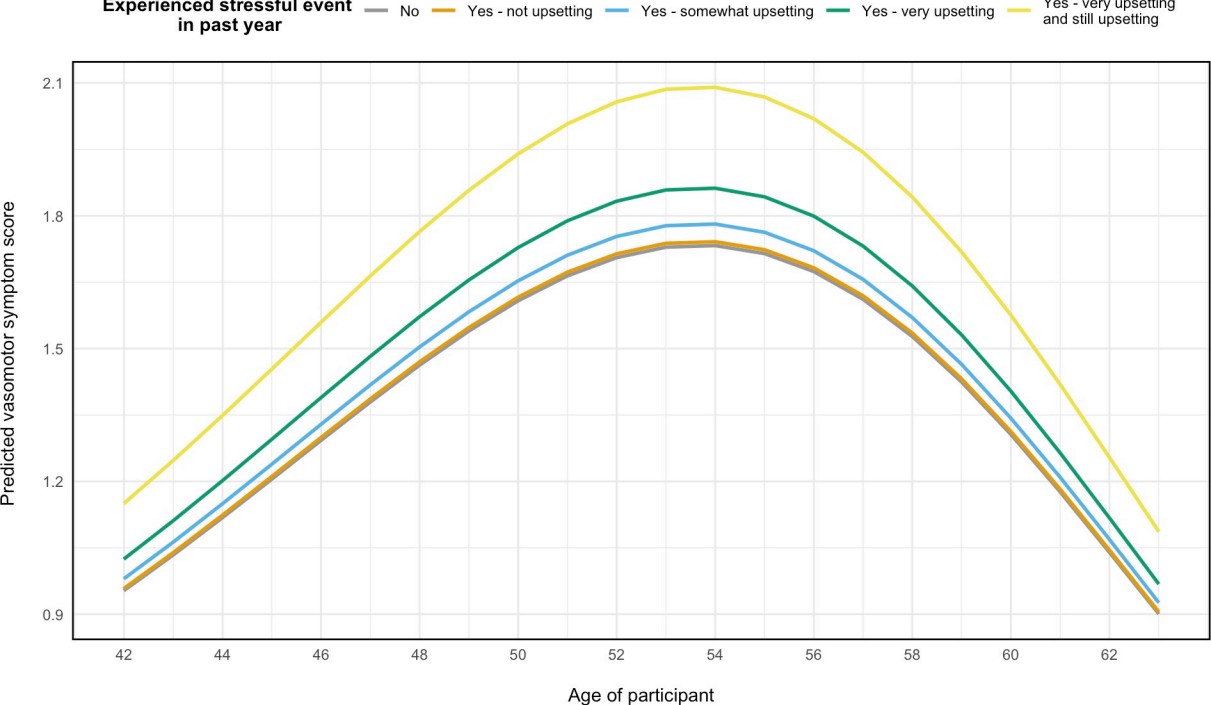

**Fig 2. Predicted vasomotor symptoms any given age based on the best fitting model.** Model adjusted for adjusted for marital status, education, smoking habits, ethnicity, self-perceived overall health, and menopause status.

**Table 4. Results from multilevel Poisson regression models showing the Incidence Rate Ratio and 95% confidence interval.**

| | Support model | Stress model 1 | Stress model 2 | Full model 1 | Full model 2 | Interaction model 1 | Interaction model 2 |
|---|---|---|---|---|---|---|---|
| Stress index (ref.: No) | | | | | | | |
| Yes—not upsetting | | 1.01 (0.95–1.06) | | 1.01 (0.95–1.06) | | 1.11 (0.78–1.60) | |
| Yes—somewhat upsetting | | 1.03 (0.98–1.07) | | 1.03 (0.98–1.07) | | 1.02 (0.79–1.30) | |
| Yes—very upsetting | | 1.07** (1.03–1.13) | | 1.07** (1.03–1.13) | | 1.31* (1.01–1.69) | |
| Yes—very upsetting and still upsetting | | 1.21*** (1.15–1.26) | | 1.20*** (1.15–1.26) | | 1.57*** (1.23–2.00) | |
| Stress dose | | | 1.03*** (1.02–1.03) | | 1.03*** (1.02–1.03) | | 1.05*** (1.02–1.07) |
| Support | 0.98 (0.96–1.00) | | | 0.99 (0.96–1.01) | 0.99 (0.96–1.01) | 1.02 (0.97–1.07) | 1.00 (0.97–1.03) |
| Social support:Stress index (ref.: No) | | | | | | | |
| Social support:Yes—not upsetting | | | | | | 0.98 (0.90–1.06) | |
| Social support:Yes— somewhat upsetting | | | | | | 1.00 (0.95–1.06) | |
| Social support:Yes–very upsetting | | | | | | 0.96 (0.90–1.01) | |
| Social support:Yes—very upsetting and still upsetting | | | | | | 0.94* (0.89–0.99) | |
| Social support:Stress dose | | | | | | | 1.00 (0.99–1.00) |

Note:

* P≤0.05

** P≤0.01

*** P≤0.001.

The dependent variable for all models is vasomotor symptoms, and all models are adjusted for age (linear, cubic and quadratic terms), marital status, smoking habits, education, ethnicity, self-perceived overall health, and menopausal status.

from a year ago is not associated with more frequent VMS (IRR: 1.00; 95% CI: 0.95–1.05), and a very upsetting and still upsetting event from a year ago being associated with a 7% increase in VMS compared to women who had experienced no stressor a year ago (95% CI: 1.01–1.12).

## Discussion

Support is often thought to buffer against the negative health impacts of stress [44], however, to our knowledge, this has not been tested with respect to menopause symptoms. It is well documented that stress associates with more severe symptoms that occur at a higher frequency [3,25–28], with some evidence that a supportive environment decreases symptoms [3,33,35,72]. In our analyses of a sample of US women, we did not find evidence that increased social support is associated with decreased menopause symptoms, nor that it moderates the negative relationship between stress and menopause symptoms. Rather, we find that a model that does not include social support or its interaction with stress best fits the data, with this model suggesting that it is not necessarily the stressful event but the degree to which it upsets the woman (i.e. perceived stress) that impacts VMS.

This is interesting for a number of reasons. These results suggest that experiencing a stressful event in and of itself does not worsen menopause symptoms; rather, it is the woman's psychological reaction to the stressful event that has the largest effect on VMS frequency. For example, based on our findings, if the period of upset after a stressful event is short, then

menopause symptoms may not be significantly impacted. This is in line with findings that perceived stress is influential over symptomology [25], and that women who are deemed to be more resilient experience fewer menopause symptoms (i.e. women who experienced a stressful event but were not upset by it) [33]. Further, as we predict that current stress influences VMS, we carried out an additional *post hoc* analysis in which we lagged the measure of stress in the best fitting model. If VMS are partially exacerbated by current stress, we would expect a lagged variable that indicates historical stress to be a poor predictor, which we found to be the case (S1 Text Table C). There was a significant effect in the lagged variable amongst women who a year ago had reported that they had experienced a life stressor that was upsetting at the time, and there are two reasons that might explain this relationship in the lagged variable: firstly, the upsetting event that happened last year could still feasibly be upsetting them now (e.g., if it was the death of a close relative); secondly, women who reported being still upset by a life stressor may be less resilient to their effects, meaning that they would be more likely to report life stressors in subsequent years. Nonetheless, these results are more evidence for a possible causal relationship between stress and VMS.

Contrary to our hypothesis, our results suggest that support may not effectively protect against the effects of stress in our sample of US women. Elsewhere in the literature, there are reports that various kinds of support (familial, emotional, etc.) are associated with fewer menopause symptoms [33,35,72], but we did not replicate these effects in our analysis. When looking at other domains of health, researchers have found that social support is not always protective against stress [73,74]. It may be in this case that the support questions were not specific enough to capture their effect on menopause symptoms (e.g. women could have been asked "how often do you have someone to talk to about your health"), and to alleviate symptoms, it could be that women need to receive targeted support [35]. Alternatively, it may be that talking about the menopause does not have a positive impact. There is evidence that the anticipation of menopause symptoms, which might occur through frequent discussion, can in fact worsen the woman's experience of such symptoms [26]. Understanding how support and discussion relate to symptom expectation is critical to understand. As the discussion of menopause becomes more normalised, so may the discourse on negative symptoms. While one could expect that normalising the menopause may ease the transition, it may in fact have the converse effect if it does cause women to expect that hot flashes and such are an inevitable negative outcome of the menopause. Thus, further research is required to understand how menopause discussion and symptoms interact with one another.

Our results also showed that women from certain ethnic backgrounds experienced worse symptoms. Here, African American women experienced the most frequent symptoms, while Chinese and Japanese Americans reported symptoms less often, which is in line with findings from previous research [3,9,13,75,76]. It has been suggested that the relationship between stress and menopause symptoms may help to explain some of this variation. In the US (where data used in this study was collected), African American individuals present more markers of stress than other ethnicities [77], likely due to economic and systematic factors [75]. However, results in this study suggest that stress does not mediate the relationship between ethnicity and VMS, with the relationship persisting after the inclusion of the stress index and stress dose within the model (S1 Text Table C). Additionally, attributing ethnic differences to stress does not account for why women from an East Asian background in this sample report fewer symptoms, as–due to being an ethnic minority–it would be predicted that due to higher levels of stress [75] they would report more symptoms than white women. Therefore, it is unlikely that stress explains the ethnic differences in menopause symptoms, and further research is still required to understand this variation.

How stress affects the menopausal transition is vital to understand, as, for many women, midlife is accompanied by many negative events such as the death of parents, children leaving home, and relationship problems [78]. The menopausal transition is also accompanied by a shift in their societal role: as fertility wanes and children move away, women experience a sense of loss of identity, which can also be highly stressful for women [79–82]. As we have shown, there is a dose-effect in regards to stress and menopause symptoms, and therefore the accumulation of emotionally stressful events can be expected to worsen the menopausal transition. Furthermore, these stressors that coincide with menopause are largely unavoidable. Familial death is an inevitability, and children will generally move away at some point. The coincidence of these events with menopause is somewhat a product of social norms associated with age of reproduction, and as a result, this is unlikely to change soon. As a means of tackling the emotional burden of midlife for women, the attitude towards ageing may require a shift. Cultures in which age is revered often associate with less severe menopause symptoms (e.g. traditional East Asian cultures) [83]; furthermore, when women report menopause as being empowering they are less likely to report symptoms [84].

Mechanistically, why stress worsens menopause symptoms is less clear. Increased levels of cortisol produced in response to stress might mimic the neuroendocrinological changes that occur alongside VMS. However, recent research has shown that increased perceived stress is not associated with higher inflammation (measured through fibrinogen), so this may not be the case [16]. Alternatively, it might be that symptoms that would have otherwise have been mild or relatively unnoticeable are exacerbated by stressors, much like how pain has been observed to be worsened by stress [85].

## Limitations

While we show a possible causal relationship between stress and VMS, further research using targeted data collection is likely to establish the precise causal pathway and mechanisms. Nonetheless, the persisting relationship between stress and menopause symptoms allows us to consider possible behavioural interventions women could use to deal with the symptoms associated with the menopausal transition. We found no relationship between our support measure and VMS. We chose to use an aggregated measure of support, which means we could not identify which measures of social support were associated with more or less frequent menopause symptoms (e.g. two women could have the same support score, but completely different support experiences). Further research may wish to look at support specific to the menopausal transition (e.g. menopause cafes), or conduct qualitative research to find which forms of support women themselves find most vital throughout menopause.

## Conclusion

This research contributes to the growing body of research linking stress with a worse menopausal transition. For most, stress is an unavoidable aspect of daily life, and our results confirm an association between stressful events and worse menopause symptoms. However, our results suggest it is not necessarily the occurrence of a stressor that exacerbates menopause symptoms, but how the woman psychologically responds to it. In our sample, women who were "still upset" by a stressful event reported the most severe symptoms, whereas women who had experienced the same event but were not upset by it did not report significantly worse symptoms. This suggests that resilience may be important in the relationship between stress and menopause symptoms; however, further research is required to establish this. Little evidence was found for support reducing menopause symptoms, or acting as a buffer against the effects of stress. This may be relevant for understanding treatments and interventions for menopause

symptoms, in particular, whether support-based treatments to tackle menopause symptoms are effective. Future research should focus on whether social interventions do have the desired effect on menopause symptoms, and also whether certain personality types–in regards to resilience–experience menopause in a different way.

## Supporting information

**S1 Text. Supporting text.** Supporting information containing information on the selection of the social support variable (Figure A), and the derivation of the stress index (Table A) and stress dose (Table B). Figure B demonstrates the justification for how missing data in the stress dose variable was dealt with, and Figure C shows the distribution of the stress dose variable by wave and age. Complete model results are presented in Table C.
(DOCX)

## Acknowledgments

The authors are grateful to the study stuff at each site and all the women who participated in SWAN. Clinical Centres: University of Michigan, Ann Arbor—Siobán Harlow, PI 2011–present, Mary Fran Sowers, PI 1994–2011; Massachusetts General Hospital, Boston, MA—Joel Finkelstein, PI 1999–present; Robert Neer, PI 1994–1999; Rush University, Rush University Medical Center, Chicago, IL—Howard Kravitz, PI 2009–present; Lynda Powell, PI 1994–2009; University of California, Davis/Kaiser—Ellen Gold, PI; University of California, Los Angeles—Gail Greendale, PI; Albert Einstein College of Medicine, Bronx, NY—Carol Derby, PI 2011–present, Rachel Wildman, PI 2010–2011; Nanette Santoro, PI 2004–2010; University of Medicine and Dentistry—New Jersey Medical School, Newark—Gerson Weiss, PI 1994–2004; and the University of Pittsburgh, Pittsburgh, PA—Karen Matthews, PI. NIH Program Office: National Institute on Aging, Bethesda, MD—Chanda Dutta 2016–present; Winifred Rossi 2012–2016; Sherry Sherman 1994–2012; Marcia Ory 1994–2001; National Institute of Nursing Research, Bethesda, MD—Program Officers. Central Laboratory: University of Michigan, Ann Arbor—Daniel McConnell (Central Ligand Assay Satellite Services). Coordinating Centre: University of Pittsburgh, Pittsburgh, PA—Maria Mori Brooks, PI 2012–present; Kim Sutton-Tyrrell, PI 2001–2012; New England Research Institutes, Watertown, MA—Sonja McKinlay, PI 1995–2001. Steering Committee: Susan Johnson, Current Chair, Chris Gallagher, Former Chair. The authors also thank one anonymous reviewer for their helpful comments.

## Author Contributions

**Conceptualization:** Megan Arnot, Ruth Mace.

**Data curation:** Megan Arnot.

**Formal analysis:** Megan Arnot.

**Methodology:** Megan Arnot.

**Supervision:** Emily H. Emmott, Ruth Mace.

**Visualization:** Megan Arnot.

**Writing – original draft:** Megan Arnot.

**Writing – review & editing:** Megan Arnot, Emily H. Emmott, Ruth Mace.

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
