## [Decision Letter · Decision Letter 0]

30 Oct 2020

PONE-D-20-28399

The relationship between social support, stressful events, and menopause symptoms

PLOS ONE

Dear Dr. Arnot,

Thank you for submitting your manuscript to PLOS ONE. After careful consideration, we feel that it has merit but does not fully meet PLOS ONE’s publication criteria as it currently stands. Therefore, we invite you to submit a revised version of the manuscript that addresses the points raised during the review process.

I know that the review has taken a long time, but is has been difficult, given the world situation, to find reviewers who accept. In general, the study is interesting, but it is necessary to attend to several theoretical and methodological issues to better support it; thus, it is important to consider the observations made to the manuscript by the reviewer in all its sections.

We look forward to receiving your revised manuscript.

Kind regards,

Martha Asuncion Sánchez-Rodríguez, PhD

Academic Editor

PLOS ONE

Additional Editor Comments:

I know that the review has taken a long time, but is has been difficult, given the world situation, to find reviewers who accept. In general, the study is interesting, but it is necessary to attend to several theoretical and methodological issues to better support it; thus, it is important to consider the observations made to the manuscript by the reviewer in all its sections.

Journal Requirements:

Reviewers' comments:

Reviewer's Responses to Questions

**Comments to the Author**

1. Is the manuscript technically sound, and do the data support the conclusions?

Reviewer #1: Partly

2. Has the statistical analysis been performed appropriately and rigorously? 

Reviewer #1: Yes

3. Have the authors made all data underlying the findings in their manuscript fully available?

Reviewer #1: Yes

4. Is the manuscript presented in an intelligible fashion and written in standard English?

Reviewer #1: Yes

5. Review Comments to the Author

Reviewer #1: This paper is focused in a very interesting study field, relationship between social support with stressful events and menopause symptoms. However, the manuscript has several weaknesses in the theoretical-support, methods and discussion. Therefore, it cannot be accepted for publication in its current form.

Major comments

1.Introduction.

• The text included in this section is too long and imprecise. In this sense, the authors should only include the theoretical support related to the study. It is suggested that you review the following articles:

(i) Thoits, P.A. Mechanisms linking social ties and support to physical and mental health. J. Health Soc. Behav. 2011, 52, 145–161. ,

(ii) Uchino, B.N., et al. The relationship between social support and physiological processes: A review with emphasis on underlying mechanisms and implications for health. Psychol. Bull. 1996, 119, 488–531.,

(iii) Uchino, B.N. Social support and health: A review of physiological processes potentially underlying links to disease outcomes. J. Behav. Med. 2006, 29, 377–387.,

(iv) Smith, K.P. and Christakis, N.A. Social networks and health. Annu. Rev. Sociol. 2008, 34, 405–429.

•It is not necessary to make the hypotheses explicit and the objective must be concise and precise.

2. Methods.

• What is the reliability of the instruments used to measure social support and stressful events? C

3. Results

• Los títulos de los cuadros deben ser menos extesos y más precisos. No es necesario que incluya • The titles of the tables should be less long and more precise. You do not need to include the reference of the statistical analysis (this was already done in the methods section).

4. Discussion.

• The authors do not analyze or compare their results with similar studies. In this sense, you must consider the similarity of the instruments used to measure the variables.

• The authors should consider that the self-report of stressful events are not necessarily related to the perception and response to stress.

•Study limitations should be included.

5. The conclusions are not based on the results.

6. PLOS authors have the option to publish the peer review history of their article (what does this mean?). If published, this will include your full peer review and any attached files.

Reviewer #1: No

---

## [Author Response · Author response to Decision Letter 0]

19 Nov 2020

We thank the reviewer for their helpful comments, and have addressed each one below.

Introduction

• The text included in this section is too long and imprecise. In this sense, the authors should only include the theoretical support related to the study. It is suggested that you review the following articles:

i. Thoits, P.A. Mechanisms linking social ties and support to physical and mental health. J. Health Soc. Behav. 2011, 52, 145–161. ,

ii. Uchino, B.N., et al. The relationship between social support and physiological processes: A review with emphasis on underlying mechanisms and implications for health. Psychol. Bull. 1996, 119, 488–531.,

iii. Uchino, B.N. Social support and health: A review of physiological processes potentially underlying links to disease outcomes. J. Behav. Med. 2006, 29, 377–387.,

iv. Smith, K.P. and Christakis, N.A. Social networks and health. Annu. Rev. Sociol. 2008, 34, 405–429

We have completely rewritten the introduction to ensure precision in language, and to not deviate from the topic at hand. We have also incorporated some of the suggested references as appropriate. 

• It is not necessary to make the hypotheses explicit and the objective must be concise and precise.

We have removed the explicit hypothesis and rewritten this paragraph:

“In this study, we build on previous research into stress, support and menopause symptoms by testing whether social support acts as an effective buffer against the negative influence of stress on VMS. First, we test whether higher levels of social support and stress associate with less frequent VMS and more frequent VMS, respectively. Further, we test whether social support buffers against the proposed negative impact of stress on VMS.”

Methods

• What is the reliability of the instruments used to measure social support and stressful events?

We had already highlighted the reliability of the measure of stressful events (L293-295): 

“This is a modified version of the Psychiatric Epidemiology Research Interview life events scale (69), which has been validated across multiple racial and ethnic groups and shows high internal consistency (70).” 

A few sentences have now been added about the reliability of our social support measure (L283-286):

“This measure is taken from the Medical Outcome Study: Social Support Survey (MOS-SSS) which is one of the most widely used instruments for measuring social support (69). The MOS-SSS has high levels of validity and reliability and has been translated into multiple languages (70-73).”

Results

• The titles of the tables should be less long and more precise. 

Thank you for this suggestion. While recognising the value of concise titles, we have decided to keep the titles of the tables as they are. This is because we are aware that readers might just look at the tables for the results/quick summary of the paper (e.g., tables and graphs are frequently shared on social media or in talks without manuscript text). Therefore, we want to ensure the tables make sense in their own right, and we have deemed the titles of the tables necessary. 

• You do not need to include the reference of the statistical analysis (this was already done in the methods section).

This has been removed. 

Discussion

• The authors do not analyze or compare their results with similar studies. In this sense, you must consider the similarity of the instruments used to measure the variables

Thank you for this suggestion. We believe we have already done this in the following sentences: 

• “This is in line with findings that perceived stress is influential over symptomology (23), and that women who are deemed to be more resilient experience fewer menopause symptoms (31).”

• “Elsewhere in the literature, there are reports that various kinds of support (familial, emotional, etc.) are associated with fewer menopause symptoms (31, 33, 70), but we did not replicate these effects in our analysis. When looking at other domains of health, researchers have found that social support is not always protective against stress (71, 72). It may be in this case that the support question was not specific enough to capture its effect on menopause symptoms (e.g. women could have been asked “how often do you have someone to talk to about your health”),”

Though, for clarity, we have added in the following sentence: 

“with previous research sometimes using combined measures of support in studies.”

• The authors should consider that the self-report of stressful events are not necessarily related to the perception and response to stress.

We are not completely clear what is meant by this comment, however, the measure of stress used in this study is self-reported and includes how upset the respondent is, which is surely subjective and therefore related to perception. For example, how upset someone is by a life stressor will be relative to past experience of stressful events, general resilience, etc. (Dohrenwend, 2006), and therefore we think that the measure of stress used in this study would be related to perception and response to stress. 

• Study limitations should be included.

Thank you for this suggestion. Limitations have been added in as follows: 

“This study is correlative, and therefore we cannot definitely say that stressful events found upsetting by a woman result in worse menopause symptoms. Further, the precise causal mechanism that might underly the relationship is not clear. Nonetheless, the persisting relationship between stress and menopause symptoms allows us to consider possible behavioural interventions women could use to deal with the symptoms associated with the menopausal transition.”

Conclusion

• The conclusions are not based on the results

Thank you for highlighting this. The conclusion has been rewritten to remove any speculative conclusions, or where appropriate we have highlighted when they are speculative inferences of results.

References

DOHRENWEND, B. P. 2006. Inventorying stressful life events as risk factors for psychopathology: Toward resolution of the problem of intracategory variability. Psychological Bulletin, 132, 477-495.

---

## [Decision Letter · Decision Letter 1]

2 Dec 2020

PONE-D-20-28399R1

The relationship between social support, stressful events, and menopause symptoms

PLOS ONE

Dear Dr. Arnot,

Thank you for submitting your manuscript to PLOS ONE. After careful consideration, we feel that it has merit but does not fully meet PLOS ONE’s publication criteria as it currently stands. Therefore, we invite you to submit a revised version of the manuscript that addresses the points raised during the review process.

This corrected version has been revised by a new reviewer to complete two reviewers,  which contributes with very important comments that may be improve the manuscript. I suggest attending to the observations, mainly those of the data analysis to better support the findings.

We look forward to receiving your revised manuscript.

Kind regards,

Martha Asuncion Sánchez-Rodríguez, PhD

Academic Editor

PLOS ONE

Additional Editor Comments (if provided):

This corrected version has been revised by a new reviewer to complete two reviewers, which contributes with very important comments that may be improve the manuscript. I suggest attending to the observations, mainly those of the data analysis to better support the findings.

Reviewers' comments:

Reviewer's Responses to Questions

**Comments to the Author**

1. If the authors have adequately addressed your comments raised in a previous round of review and you feel that this manuscript is now acceptable for publication, you may indicate that here to bypass the “Comments to the Author” section, enter your conflict of interest statement in the “Confidential to Editor” section, and submit your "Accept" recommendation.

Reviewer #1: All comments have been addressed

Reviewer #2: (No Response)

2. Is the manuscript technically sound, and do the data support the conclusions?

Reviewer #1: Yes

Reviewer #2: Partly

3. Has the statistical analysis been performed appropriately and rigorously? 

Reviewer #1: Yes

Reviewer #2: No

4. Have the authors made all data underlying the findings in their manuscript fully available?

Reviewer #1: Yes

Reviewer #2: (No Response)

5. Is the manuscript presented in an intelligible fashion and written in standard English?

Reviewer #1: Yes

Reviewer #2: Yes

6. Review Comments to the Author

Reviewer #1: In general, the authors have made corrections to the manuscript considering the comments. However, I reiterate that it is necessary that the titles of tables 2 and 3 should be more concise. In this regard, the specifications can be included at the bottom of the table.

Reviewer #2: The authors are seeking to examine support buffering hypothesis in stress and menopause symptoms. I did not review the original submission – so I am responding to the revisions that were requested and made and the revised manuscript. The topic is interesting and important and the authors use a wonderful dataset to conduct their analyses. However, I think that the authors stop short in their analyses to actually advance the research in a substantial way.

First, the Introduction is well written – so no major issues there and I think that the authors addressed the prior reviewers’ comments adequately. However, the final sentence in the Introduction threw me. The authors state: “This research aims to clarify how support and stress are associated with one another in regards to VMS, as well as informing the potential efficacy of support-based treatments – such as mindfulness and mediation – to manage menopause symptoms.” This makes it sound like the authors are going to be looking at mindfulness and meditation (I think they meant this instead of “mediation”) in their analyses. However, in reading the rest of the manuscript, there is nothing about these concepts in the analyses. Also, this is the first mention of these constructs in the Introduction. I suggest that this sentence be removed and maybe moved to the Discussion as a future direction.

With respect to the social support measure, I am concerned about the authors letting the data drive their analyses. In my reading, they first tested which social support measures best predicted VMS and then chose the one that best predicted VMS to go forward with in their analyses. First, this feels like data mining to me. Second, if social support is a moderator, there is no reason that it needs to be related to the outcome (see Baron & Kenny, 1986) – in other words, a moderator is supposed to be orthogonal to the predictor and outcome.

Additionally, the authors highlight the race/ethnic differences in levels VMS symptoms and then speculate what that might mean. But, they have the data to conduct analyses to see if the link between stress and VMS differs by race/ethnicity and whether the support buffer is found for some race/ethnic groups as opposed to others.

Finally, they mention that a limitation of their analyses is that they are correlative. However, they have multiple waves of data and could easily have conducted lagged analyses to determine whether causation would be supported.

Overall, I feel that the authors need to do more in their analyses and that as of now, they essentially only have shown that stress is connected with VMS symptoms and I am not sure that is substantive enough for publication. They have the data to do a lot more – and it should not be too hard to do these additional analyses.

Minor Points:

In the first paragraph, first sentence, the authors say “The menopause is often a significant event for women…” I find the use of “The” to be awkward – I think it should just say “Menopause is often a significant event for women…”

In Table 2, I think there is a typo – half the sample is Chinese American and only 7% white but that does not match with the text. Also, Caucasian is no longer used (it is actually attached to white supremacists: https://www.latimes.com/opinion/story/2019-09-10/race-caucasian-myth-racism) – instead, it should be listed as “White”.

7. PLOS authors have the option to publish the peer review history of their article (what does this mean?). If published, this will include your full peer review and any attached files.

Reviewer #1: **Yes: **Víctor Manuel Mendoza-Núñez

Reviewer #2: No

---

## [Author Response · Author response to Decision Letter 1]

14 Dec 2020

We would like to thank the reviewers for their helpful comments. We have addressed each one below.

Reviewer #1

• In general, the authors have made corrections to the manuscript considering the comments. However, I reiterate that it is necessary that the titles of tables 2 and 3 should be more concise. In this regard, the specifications can be included at the bottom of the table.

Excess information has been moved to the bottom of the table. 

Reviewer #2

• The authors state: “This research aims to clarify how support and stress are associated with one another in regards to VMS, as well as informing the potential efficacy of support-based treatments – such as mindfulness and mediation – to manage menopause symptoms.” This makes it sound like the authors are going to be looking at mindfulness and meditation (I think they meant this instead of “mediation”) in their analyses. However, in reading the rest of the manuscript, there is nothing about these concepts in the analyses. Also, this is the first mention of these constructs in the Introduction. I suggest that this sentence be removed and maybe moved to the Discussion as a future direction.

This sentence has been removed from the introduction. 

• With respect to the social support measure, I am concerned about the authors letting the data drive their analyses. In my reading, they first tested which social support measures best predicted VMS and then chose the one that best predicted VMS to go forward with in their analyses. First, this feels like data mining to me. Second, if social support is a moderator, there is no reason that it needs to be related to the outcome (see Baron & Kenny, 1986) – in other words, a moderator is supposed to be orthogonal to the predictor and outcome.

Thank you for pointing this out. We were by no means trying to data mine. Rather, as it’s survey data with hundreds of possible variables, we were using pre-specified variable selection strategies. However, we have taken this comment on board and changed the measure of social support used in the analysis, and instead used a composite measure that includes the four forms of support collected by SWAN. This has not changed the overall results of the analysis, and we have included the individual measures in the S1 Text. 

• Additionally, the authors highlight the race/ethnic differences in levels VMS symptoms and then speculate what that might mean. But, they have the data to conduct analyses to see if the link between stress and VMS differs by race/ethnicity and whether the support buffer is found for some race/ethnic groups as opposed to others.

Thank you for highlighting this. We don’t wish to specifically add this as a research question as it is not one of our aims; however, we do now mention it more specifically in the results and comment on how our data fit the claims that stress underlies the relationship between ethnicity and VMS: “Here, African American women experienced the most frequent symptoms, while Chinese and Japanese Americans reported symptoms less often, which is in line with findings from previous research (3, 9, 13, 75, 76). It has been suggested that the relationship between stress and menopause symptoms may help to explain some of this variation. In the US (where data used in this study was collected), African American individuals present more markers of stress than other ethnicities (77), likely due to economic and systematic factors (75). However, results in this study suggest that stress does not mediate the relationship between ethnicity and VMS, with the relationship persisting after the inclusion of the stress index and stress dose within the model (S1 Text Table C). Additionally, attributing ethnic differences to stress does not account for why women from an East Asian background in this sample report fewer symptoms, as – due to being an ethnic minority – it would be predicted that due to higher levels of stress (75) they would report more symptoms than white women. Therefore, it is unlikely that stress explains the ethnic differences in menopause symptoms, and further research is still required to understand this variation.”

• Finally, they mention that a limitation of their analyses is that they are correlative. However, they have multiple waves of data and could easily have conducted lagged analyses to determine whether causation would be supported.

Thank you for bringing this up. We do not necessarily think a lagged analysis would be theoretically appropriate: as we explain in the introduction, we are suggesting that current stress is driving the relationship between stress and VMS, and therefore we would likely need data on a shorter timescale to establish a definitive causal relationship. Further, by conducting a lagged analysis, we are still only really showing a correlation between stress at t-1 and VMS, and so a significant effect could be found by chance. However, we have added in a post hoc analysis into the results section and discussion, were we find that current stress better predicts VMS than historic stress, which does support our notion that there is a causal relationship. 

• In the first paragraph, first sentence, the authors say “The menopause is often a significant event for women…” I find the use of “The” to be awkward – I think it should just say “Menopause is often a significant event for women…”

This has been changed. 

• In Table 2, I think there is a typo – half the sample is Chinese American and only 7% white but that does not match with the text. Also, Caucasian is no longer used (it is actually attached to white supremacists: https://www.latimes.com/opinion/story/2019-09-10/race-caucasian-myth-racism) – instead, it should be listed as “White”.

Thanks for pulling these issues up – they have been changed accordingly.

---

## [Decision Letter · Decision Letter 2]

2 Jan 2021

The relationship between social support, stressful events, and menopause symptoms

PONE-D-20-28399R2

Dear Dr. Arnot,

We’re pleased to inform you that your manuscript has been judged scientifically suitable for publication and will be formally accepted for publication once it meets all outstanding technical requirements.

Kind regards,

Martha Asuncion Sánchez-Rodríguez, PhD

Academic Editor

PLOS ONE

Additional Editor Comments (optional):

The reviews and I agree that it is an interesting paper, congratulations and happy new year.

Reviewers' comments:

Reviewer's Responses to Questions

**Comments to the Author**

1. If the authors have adequately addressed your comments raised in a previous round of review and you feel that this manuscript is now acceptable for publication, you may indicate that here to bypass the “Comments to the Author” section, enter your conflict of interest statement in the “Confidential to Editor” section, and submit your "Accept" recommendation.

Reviewer #1: All comments have been addressed

Reviewer #2: All comments have been addressed

2. Is the manuscript technically sound, and do the data support the conclusions?

Reviewer #1: Yes

Reviewer #2: Yes

3. Has the statistical analysis been performed appropriately and rigorously? 

Reviewer #1: Yes

Reviewer #2: Yes

4. Have the authors made all data underlying the findings in their manuscript fully available?

Reviewer #1: Yes

Reviewer #2: Yes

5. Is the manuscript presented in an intelligible fashion and written in standard English?

Reviewer #1: Yes

Reviewer #2: Yes

6. Review Comments to the Author

Reviewer #1: This paper is focused in a very interesting study field, relationship between social support with stressful events and menopause symptoms.

The authors have corrected the manuscript considering all the comments.

Reviewer #2: I feel that the authors have addressed all of my concerns. I especially think it is interesting that the lagged analyses provided more support for their argument about current stress vs. historic stress. I still think it would be interesting to include race/ethnicity as a moderator but I understand that it was not a main aim of their study. Nice job and interesting paper.

7. PLOS authors have the option to publish the peer review history of their article (what does this mean?). If published, this will include your full peer review and any attached files.

Reviewer #1: **Yes: **Víctor Manuel Mendoza-Núñez PhD

Reviewer #2: No

---

## [Editor Report · Acceptance letter]

6 Jan 2021

PONE-D-20-28399R2 

The relationship between social support, stressful events, and menopause symptoms 

Dear Dr. Arnot:

I'm pleased to inform you that your manuscript has been deemed suitable for publication in PLOS ONE. Congratulations! Your manuscript is now with our production department. 

Kind regards, 

on behalf of

Dr. Martha Asuncion Sánchez-Rodríguez 

Academic Editor

PLOS ONE